# Redistribution of fragmented mitochondria ensures symmetric organelle partitioning and faithful chromosome segregation in mitotic mouse zygotes

Haruna Gekko[1†], Ruri Nomura[1†], Daiki Kuzuhara[2], Masato Kaneyasu[1], Genpei Koseki[1], Deepak Adhikari[3], Yasuyuki Mio[2], John Carroll[3], Tomohiro Kono[4], Hiroaki Funahashi[1], Takuya Wakai[1*]

[1]Department of Animal Science, Graduate School of Environment and Life Science, Okayama University, Okayama, Japan; [2]Reproductive Centre, Mio Fertility Clinic, Tottori, Japan; [3]Development and Stem Cell Program and Department of Anatomy and Developmental Biology, Monash Biomedicine Discovery Institute, Monash University, Melbourne, Australia; [4]Department of Bioscience, Tokyo University of Agriculture, Tokyo, Japan

*For correspondence: t2wakai@okayama-u.ac.jp

†These authors contributed equally to this work

Competing interest: The authors declare that no competing interests exist.

## eLife Assessment

This **important** study investigates the role of Drp1 in early embryo development. The authors have addressed most of the original comments and the work now presents **convincing** evidence on how this protein influences mitochondrial localization and partitioning during the first embryonic divisions. The research employs the Trim-Away technique to eliminate Drp1 in zygotes, revealing critical insights into mitochondrial clustering, spindle formation, and embryonic development.

**Abstract** In cleavage-stage embryos, preexisting organelles partition evenly into daughter blastomeres without significant cell growth after symmetric cell division. The presence of mitochondrial DNA within mitochondria and its restricted replication during preimplantation development makes their inheritance particularly important. While chromosomes are precisely segregated by the mitotic spindle, the mechanisms controlling mitochondrial partitioning remain poorly understood. In this study, we investigate the mechanism by which Dynamin-related protein 1 (Drp1) controls the mitochondrial redistribution and partitioning during embryonic cleavage. Depletion of Drp1 in mouse zygotes causes marked mitochondrial aggregation, and the majority of embryos arrest at the 2 cell stage. Clumped mitochondria are located in the center of mitotic Drp1-depleted zygotes with less uniform distribution, thereby preventing their symmetric partitioning. Asymmetric mitochondrial inheritance is accompanied by functionally inequivalent blastomeres with biased ATP and endoplasmic reticulum $Ca^{2+}$ levels. We also find that marked mitochondrial centration in Drp1-depleted zygotes prevents the assembly of parental chromosomes, resulting in chromosome segregation defects and binucleation. Thus, mitochondrial fragmentation mediated by Drp1 ensures proper organelle positioning and partitioning into functional daughters during the first embryonic cleavage.

## Introduction

Preimplantation development of mammalian embryos consists of a series of symmetric cell divisions without significant cell growth, known as embryonic cleavage, resulting in approximately half of the cytoplasm in each resulting cell, called a blastomere. The segregation of maternal content between daughter blastomeres allows rapid division of a single-cell zygote into multicellular organisms. To ensure the correct segregation of cellular content to produce functional daughters, cells must duplicate and apportion their various organelles with high accuracy (*Carlton et al., 2020*). Chromosomes are assembled and segregated into daughter cells with the formation of mitotic spindles, whereas membrane-bound organelles, such as the endoplasmic reticulum (ER) and mitochondria, are typically generated from existing structures, and both daughters receive a share of these components during cell division. This is particularly important in the case of mitochondria because mitochondria are semi-autonomous organelles that contain their own genome, mitochondrial DNA (mtDNA). Growing oocytes exponentially increase their mtDNA, and a fully grown oocyte contains more than 200 000 copies of mtDNA (*Mahrous et al., 2012*). After fertilization, abundant preexisting mtDNA is segregated among daughter cells without replication during preimplantation development (*Poulton et al., 2010*). As mitochondria are the primary source of ATP production via oxidative phosphorylation in oocytes and early embryos owing to low glycolytic activity (*Dumollard et al., 2007*), proper inheritance must be ensured for cell energetic homogeneity among blastomeres. Mitochondria also play an important role in the regulation of intracellular $Ca^{2+}$ signaling, which in turn controls cell metabolism and cell death pathways (*Rizzuto et al., 2012*). Thus, defects in mitochondrial inheritance are likely to have serious consequences for daughter blastomeres and manifest as the failure of pre- or post-implantation development.

Mitochondria are highly dynamic organelles that move along cytoskeletal networks and undergo continuous fusion and fission; these dynamics allow cells to respond and adapt to various intracellular and extracellular changes and maintain cellular function and homeostasis (*Mishra and Chan, 2014*; *Westermann, 2010*). Active partitioning of mitochondria is observed in budding yeast fission, which undergoes asymmetric cell division; mitochondria actively move along actin filaments into the daughter cell (*Mishra and Chan, 2014*). In mammalian cells that divide symmetrically, the partitioning of mitochondria is largely due to passive processes, as fragmented and dispersed mitochondrial distribution allows for stochastic and roughly equal partitioning into daughter cells. Mitochondrial fission is thus important for faithful inheritance and proper intracellular distribution of the mitochondria (*Westermann, 2010*). Whether active transport of mitochondria by the cytoskeleton contributes to mitochondrial inheritance in dividing animal cells remains controversial. Microtubules function as cytoskeletal platforms for the distribution and transport of mitochondria. During mitosis, mitochondria decouple from spindle microtubules and disperse in the cell periphery, allowing for passive segregation (*Chung et al., 2016*). Furthermore, recent studies have shown that actin filaments take over as the dominant mitochondrial scaffold following mitochondrial release from microtubules at the start of mitosis (*Moore et al., 2021*). Myosin 19 (Myo19), a mitochondrial-localized myosin in vertebrates, acts as an actin-based motor for mitochondrial dynamics. Notably, depletion of Myo19 causes asymmetrical mitochondrial inheritance between daughter cells (*Rohn et al., 2014*), though the molecular characteristics of Myo19 in mammalian oocytes/eggs have yet to be shown.

Dynamin-related protein 1 (Drp1) is a key regulator of mitochondrial fission in many eukaryotic organisms. Drp1 is recruited to the mitochondrial membrane where it forms helical oligomers that induce membrane constriction and severing (*Westermann, 2010*). The indispensable role of Drp1-mediated mitochondrial fission in mammalian oocytes has been documented using conditional knockout (KO) mice (*Udagawa et al., 2014*). Drp1 KO oocytes undergo meiotic arrest due to severe organelle aggregation, which complicates the functional analysis of Drp1 in mature oocytes and embryos. More recently, mature oocytes have been successfully isolated from juvenile Drp1 conditional KO mice (*Adhikari et al., 2022*). After parthenogenetic activation of Drp1 KO oocytes, few embryos develop to the blastocyst stage. Nevertheless, the precise role of mitochondrial fission in normal embryos remains elusive. To this end, we employed Trim-Away, a method to degrade endogenous proteins via the proteasome-mediated pathway (*Clift et al., 2017*), which acutely depletes large amounts of storage proteins in the oocyte, allowing loss-of-function studies during the preimplantation development from the zygotic stage. In this study, we show that Drp1 depletion in fertilized zygotes causes marked mitochondrial aggregation and early embryonic arrest. Live imaging of mitochondria

during the first cleavage division reveals that loss of Drp1 disturbs the spatiotemporal mitochondrial dynamics, resulting in an asymmetric mitochondrial inheritance between daughter blastomeres with functional heterogeneities. We also find that misplaced mitochondria impair the assembly of parental spindles, leading to binucleated blastomeres, which closely resemble a clinical phenotype of human embryos in in vitro fertilization (IVF) procedures (*Hardy et al., 1993*; *Meriano et al., 2004*; *Seikkula et al., 2018*).

## Results

### Fragmented mitochondria are redistributed during the first embryonic cleavage and equally partitioned into daughter blastomeres

We first imaged mitochondria and chromosomes in zygotes expressing mitochondrially-targeted GFP (mt-GFP) and histone H2B-mCherry (*Figure 1A*; *Figure 1—video 1*). Mitochondria in interphase zygotes (27 hr post-hCG) progressively accumulated around the two pronuclei (*Figure 1B*, left), encircled metaphase (31 hr post-hCG) chromosomes after nuclear envelope breakdown (NEB, 30 hr post-hCG). During the transition from anaphase to telophase (32 hr post-hCG), the cytokinetic furrow constricts the mitochondrial ring that effectively divides the mitochondria approximately equally into the two daughter blastomeres, and mitochondria were dispersed throughout the cytoplasm in the interphase 2 cell embryos (35 hr post-hCG) (*Figure 1B*, right). Immunofluorescence staining of microtubules also showed mitochondria progressively surround the spindle and disperse back into the cytoplasm after the first cleavage (*Figure 1—figure supplement 1A*). The accumulation of mitochondria around the spindle is unique to the first cleavage division (*Figure 1—figure supplement 1B and C*), as live imaging of fluorescently labeled microtubules (EB3-GFP) and mitochondria (mt-DsRed) confirmed spindle-associated accumulation of mitochondria in one-cell zygotes, but no evidence of mitochondrial accumulation was observed in the spindles of 2 cell or 4 cell stage embryos. By electron microscopy (EM), we found that these accumulated mitochondria around the metaphase spindle were highly fragmented (*Figure 1C*). The ERs were also abundantly distributed around the spindle and were in partial contact with the mitochondria, and enriched bundles of actin filaments were found around these organelles.

Interphase zygotes stained with fluorescence-labelled phalloidin displayed a mitochondrial distribution pattern closely associated with F-actin cytoplasmic meshwork (*Figure 1D*; *Figure 1—figure supplement 1D*). This cytoplasmic meshwork was subsequently reorganized around the spindle at the metaphase and increased co-localization with mitochondria (*Figure 1E*). In interphase 2 cell stage embryos, the F-actin meshwork was seen again, albeit with significantly reduced overlap with mitochondria. To test whether the F-actin is involved in mitochondrial distribution and partitioning, zygotes were treated with 1 µM latrunculin A for 1 hr just after NEB, sufficient for F-actin depolymerization (*Figure 1—figure supplement 1E*). Disruption of F-actin organization resulted in asymmetric mitochondrial inheritance (*Figure 1—figure supplement 1F and G*), but also increased cell size asymmetry (*Figure 1—figure supplement 1H*), and these two asymmetries were correlated (*Figure 1—figure supplement 1I*). Given many cellular roles, disruption of F-actin per se was unsuitable as a strategy for manipulating mitochondrial distribution. We have, therefore, targeted Myo19 as a strategy for altering the mitochondrial distribution and asked if it is critical for the early development. Asymmetric mitochondrial inheritance has been previously reported in cells lacking Myo19 (*Majstrowicz et al., 2021*; *Moore et al., 2021*; *Rohn et al., 2014*). Myo19 was expressed in both interphase and metaphase zygotes, where its localization was consistent with the mitochondrial distribution (*Figure 1F*). Acute depletion of Myo19 at the PN stage by Trim-Away was confirmed by Western blot analysis, which was not observed in control zygotes injected with Trim21 mRNA and control IgG (*Figure 1G and H*). Unexpectedly, no significant asymmetry was observed in the mitochondrial distribution and partitioning between the daughter blastomeres in Myo19-depleted 2 cell embryos (*Figure 1I and J*). Live imaging of mitochondria (mt-GFP) and chromosomes (H2B-mCherry) in Myo19-depleted zygotes shows symmetric distribution and partitioning of mitochondria during the first embryonic cleavage (*Figure 1—figure supplement 2A and B*; *Figure 1—video 2*). Moreover, Myo19 depletion did not affect developmental competence of the embryos to the blastocyst stage (*Figure 1K*). Overall, fragmented mitochondria are closely associated with cytoplasmic F-actin during the first cleavage, but active transport by the actin motor is not directly involved in mitochondrial partitioning.

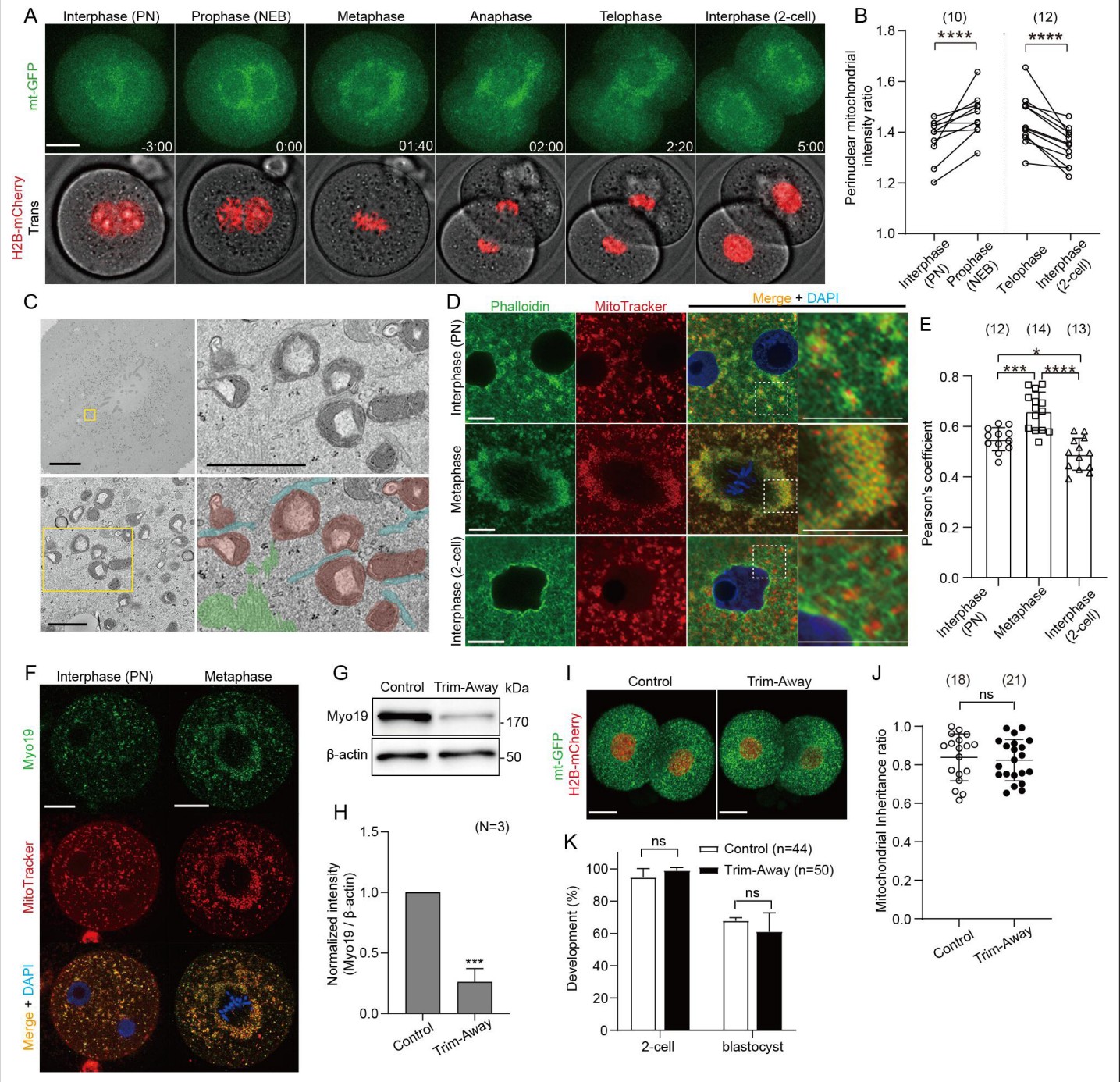

**Figure 1.** Redistribution and symmetric partitioning of mitochondria during embryonic cleavage. (**A**) Representative time-lapse images (maximum-intensity Z projection) of mitochondria and chromosomes in zygotes expressing mitochondrially-targeted (mt-GFP) and histone H2B-mCherry during the first cleavage division. Time is relative to the onset of NEBD. Scale bar, 20 μm. (**B**) Quantification of mitochondrial accumulation around the nucleus at different cell cycle phases by calculating the ratio of the mitochondrial fluorescence intensity at the nuclear periphery to that of outside the nuclear periphery. (**C**) Representative electron microscopy (EM) images (n=5) showing the accumulation of mitochondria around the mitotic spindle. Higher-magnification images (31 sections) of the representative boxed areas in the lower left and upper right, respectively. Mitochondria (purple), endoplasmic reticulum (aqua), and actin filaments (yellow-green) are pseudo-colored in the lower right panel. Scale bar left: 10 μm; Lower left and upper right: 1 μm. (**D**) Interphase or metaphase zygotes and interphase 2 cell embryos were stained with Phalloidin-iFluor 488, MitoTracker Red CMXros, and Hoechst 33342. Magnified images in the box areas are shown in the right panels. Scale bar, 10 μm. An overall view of each embryo is shown in *Figure 1—figure supplement 1D*. (**E**) Quantification of the correlation between the localization of the F-actin (Phalloidin-iFluor 488) and mitochondria (MitoTracker Red CMXros). (**F**) Representative immunofluorescence images of Myo19. Interphase (n=13) and metaphase (n=11) zygotes were stained with MitoTracker

*Figure 1 continued on next page*

*Figure 1 continued*

Red CMXRos. After fixation and immunostaining with anti-Myo19 antibody and Hoechst 33342, single-section images crossing the mid-zone of the zygotes were imaged by confocal microscopy. Scale bar, 20 μm. (**G**) Western blot analysis of zygotes overexpressing Trim21 and microinjected with control IgG or anti-Myo19 antibodies. Lysates of 100 zygotes were extracted 5 h after microinjection and were probed with antibodies specific to Myo19 and β-actin. (**H**) Quantification of the relative Myo19 expression levels in (**G**) across three experimental replications following Myo19 depletion. (**J**) Quantitation of mitochondrial mass inheritance after the first cleavage division of control and Myo19-depleted zygotes. Total mitochondrial fluorescence in each daughter blastomere of 2 cell embryos was measured, and the smaller value was divided by the greater value for the inheritance ratio. (**K**) Developmental competence of control and Myo19-depleted embryos. Percentage of zygotes reaching the indicated developmental stage. (**I**) Maximum-intensity Z projection of control and Myo19 Trim-Away embryos expressing mt-GFP and H2B-mCherry. Scale bar, 20 μm. Data are represented as mean ± SD and p-values calculated using two-tailed paired (**B**) or unpaired (**E, H, K, and J**) Student's t test. *p<0.05, ***p<0.001, ****p<0.0001; ns, not significant. Number of zygotes/embryos is indicated in brackets.

The online version of this article includes the following video, source data, and figure supplement(s) for figure 1:

**Source data 1.** Original files of the full raw uncropped blots displayed in *Figure 1G*.

**Source data 2.** Figures with the uncropped blots with the relevant bands and treatments for *Figure 1G*.

**Figure supplement 1.** Redistribution of mitochondria during the first embryonic cleavage.

**Figure supplement 2.** Redistribution of mitochondria during the first cleavage in Myo19-depleted zygotes.

**Figure 1—video 1.** Symmetric mitochondrial partitioning during the first cleavage, related to *Figure 1*.

https://elifesciences.org/articles/99936/figures#fig1video1

**Figure 1—video 2.** Mitochondrial dynamics during the first embryonic cleavage in Myo19-depleted zygotes, related to *Figure 1—figure supplement 2*.

https://elifesciences.org/articles/99936/figures#fig1video2

## Loss of Drp1 induces mitochondrial aggregation and disturbs subcellular organelle compartments

The highly fragmented mitochondrial morphology appears to facilitate mitochondrial partitioning during embryonic cleavage. To understand the underlying mechanism of this process, we examined the role of Drp1, a key regulator of mitochondrial fission. Western blot analysis revealed that Drp1 was expressed in PN zygotes, and comparable protein levels were found in the morula stage, whereas their expression was remarkably decreased at the blastocyst stage (*Figure 2A and B*). To demonstrate the role of Drp1-mediated mitochondrial fission in preimplantation embryos, we employed Trim-Away. Co-injection of *Trim21* mRNA and an antibody against Drp1 into zygotes 20–22 hr post hCG reduced the Drp1 protein to nearly undetectable levels by 5 hr post-injection at the PN stage, whereas co-injection of *Trim21* mRNA and control IgG failed to trigger protein degradation (*Figure 2C and D*).

Mitochondria in live 2 cell embryos (37/37) expressing mt-GFP were interspersed throughout the cytoplasm. In contrast, almost all Drp1 Trim-Away embryos (55/57) exhibited marked mitochondrial aggregation that was reversed by ectopic expression of mCherry-tagged Drp1 (mCh-Drp1) (50/53) (*Figure 2E and F*). In Drp1-depleted embryos, mitochondria became highly aggregated into a small number of large clusters, whereas in controls and Drp1 Trim-Away embryos expressing mCh-Drp1, mitochondria remained dispersed in a large number of small clusters (*Figure 2G and H*). We observed that similar mitochondrial aggregation occurred in parthenogenetic Drp1 KO embryos (*Figure 2—figure supplement 1A*), as the mitochondrial cluster size in $Drp1^{\Delta/\Delta}$ parthenotes was greater than $Drp1^{fl/fl}$ control, albeit the number of clusters were comparable (*Figure 2—figure supplement 1B and C*).

EM images of control 2 cell embryos showed that rounded mitochondria with less developed cristae were dispersed throughout the cytoplasm, whereas swollen or partially elongated mitochondria with lamella cristae structures in the inner membrane were observed in Drp1-depleted embryos (*Figure 2I*). Both mitochondrial major and minor axes in Drp1-depleted embryos was significantly greater than those of control embryos (*Figure 2J–L*). The mean aspect ratio (major axis/minor axis) increased slightly from 1.36 in control to 1.66 in Drp1-depleted embryos (*Figure 2M*).

To clarify the effects of these morphological changes in mitochondria on their energy production, we compared ATP levels between control and Drp1 Trim-Away embryos at the 2 cell stage. To analyze intracellular ATP concentrations, we expressed a fluorescence resonance energy transfer (FRET)-based ATP biosensor, ATeam AT1.03, with an emission ratio of AT1.03 fluorescence (YFP/CFP) used to estimate ATP levels. ATP levels in Drp1-depleted embryos were indistinguishable from those

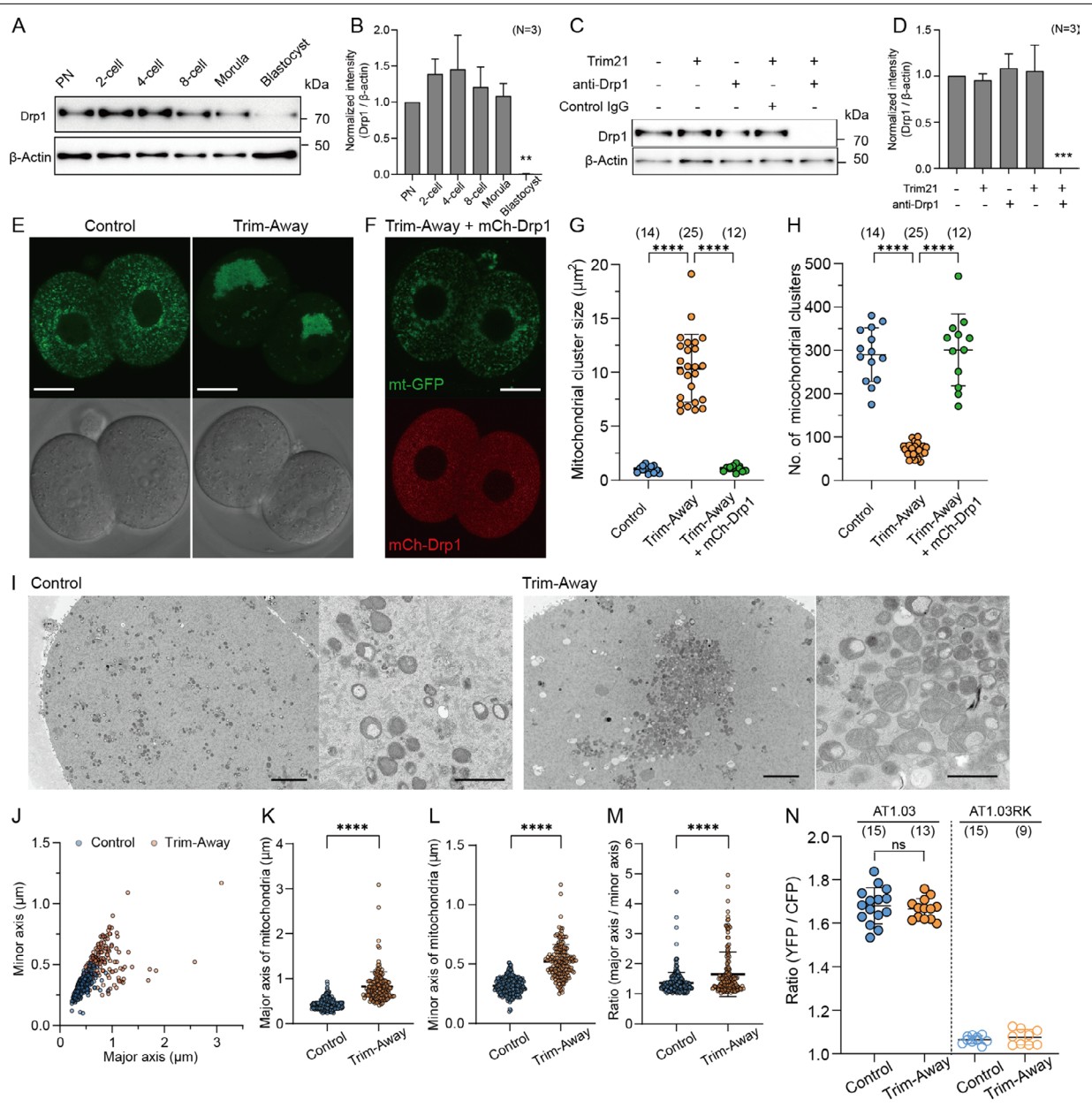

**Figure 2.** Depletion of Drp1 induces mitochondrial aggregation. (**A**) Western blot analysis of preimplantation embryos at the different stages. Lysates of 30 cells were probed with antibodies specific to Drp1 and β-actin. (**B**) Quantification of the relative Drp1 expression levels in (**A**) across three experimental replications. (**C**) Western blot analysis of zygotes overexpressing Trim21 and microinjected with control IgG or anti-Drp1 antibodies (henceforth referred to simply as control and Trim-Away, respectively), or zygotes microinjected with Trim21 mRNA, control IgG, or anti-Drp1 alone. Lysates were extracted 5 h after microinjection and immunoblotted for the indicated proteins. (**D**) Quantification of the relative Drp1 expression levels in (**C**) across three experimental replications following Drp1 depletion. (**E**) Representative images of in-control and Drp1 Trim-Away embryos expressing mt-GFP. Scale bar, 20 μm. (**F**) A representative image of mitochondria (mt-GFP) in Drp1-depleted embryos expressing exogenous Drp1 (mCh-Drp1). Note that mCh-Drp1 expression rescues the mitochondrial aggregation following Drp1 depletion by Trim-Away to the normal dispersed distribution. Scale bar, 20 μm. (**G, H**) Comparisons of size (**G**) and number (**H**) of mitochondrial clusters (mt-GFP) in (**E**) and (**F**). (**I**) Representative EM images of control (n=5, 17 sections) and Drp1-depleted (n=6, 20 sections) 2 cell embryos. Right panels show higher-magnification images. Scale bar left: 4 μm; right: 1 μm. (**J–M**) Quantification of major and minor axes (**J**) of individual mitochondria observed in (**I**). Comparisons of major axis (**K**), minor axis (**L**), and the aspect ratio (**M**) of mitochondria in control and Drp1-depleted embryos. (**N**) Levels of ATP in control and Drp1-depleted embryos were estimated using the emission ratio of AT1.03 (YFP/CFP). The emission ratio of AT1.03RK (YFP/CFP), which is unable to bind ATP, is shown as a negative control. Data are represented as mean ± SD and p-values calculated using two-tailed Student's t-test with Welch's correction. ****p<0.0001. Number of embryos are indicated in brackets.

The online version of this article includes the following source data and figure supplement(s) for figure 2:

*Figure 2 continued on next page*

*Figure 2 continued*

**Source data 1.** Original files of the full raw uncropped blots displayed in *Figures 2A and C*.

**Source data 2.** Figures with the uncropped blots with the relevant bands and treatments for *Figures 2A and C*.

**Figure supplement 1.** Mitochondrial aggregation in Drp1 knockout (KO) parthenotes.

**Figure supplement 2.** Mitochondrial aggregation disturbs subcellular organelle compartments.

in the control embryos (*Figure 2N*). No FRET signal was detected in embryos expressing a mutant version of the ATP probe AT1.03RK that cannot bind ATP.

Recent studies have focused on the interactions between organelles at membrane contact sites (MCSs) and their roles in maintaining cellular homeostasis (*English and Voeltz, 2013*). The ER is a continuous membrane-bound organelle with MCSs such as with the plasma membrane, mitochondria, Golgi, and peroxisomes. Since mitochondrial dynamics are spatially coordinated at the ER-mitochondria MCSs (*Friedman et al., 2011*), we surmised that mitochondrial aggregation due to the loss of Drp1 would compromise the organization of ER MCSs. To visualize the distribution of mitochondria with other organelles in live embryos, fluorescently tagged ER (ER-mCherry), Golgi (Golgi-mCherry), and peroxisomes (PEX-mCherry) were co-expressed with mt-GFP. The ER was partially confined to the regions of the mitochondrial aggregation in Drp1 Trim-Away embryos, which appeared to disturb the endogenous ER network (*Figure 2—figure supplement 2A*). Linscan of mt-GFP and ER-mCherry signals at each focal plane of large mitochondrial clumps showed that ER is confined to mitochondrial aggregation. In contrast, the subcellular distribution of the Golgi was not clearly altered by Drp1 depletion (*Figure 2—figure supplement 2B*). Drp1 is also known to regulate peroxisomal fission, as elongated peroxisomes are seen in Drp1-deficient cells (*Koch et al., 2003*) however, the subcellular distribution of peroxisomes in mammalian embryos has not been reported. The peroxisomes in normal 2 cell embryos showed a punctate distribution in the cytoplasm, whereas these puncta partially aggregated following Drp1 depletion by Trim-Away (*Figure 2—figure supplement 2C*), although it appeared to be independent of mitochondrial aggregation in terms of spatial positioning.

## Drp1-mediated mitochondrial fragmentation is required for the symmetric mitochondrial partitioning into two functional daughter blastomeres

To decipher the spatiotemporal regulation of mitochondrial dynamics following Drp1 depletion, we imaged mitochondria and chromosomes in Drp1 Trim-Away zygotes expressing mt-GFP and histone H2B-mCherry (*Figure 3A*; *Figure 3—video 1*). No significant delay in cell cycle progression was observed following Drp1 depletion (data not shown) compared to control zygotes (*Figure 1A*). Clumped mitochondria were located in the center of the metaphase zygotes, which resulted in an increase in mitochondrial asymmetry at anaphase (*Figure 3B*). We postulated that this asymmetry was due to non-uniformity in the distribution of mitochondria around the spindle. As expected, live imaging of mitochondria (mt-DsRed) and spindle microtubules (EB3-GFP) revealed compared with the uniform angular positioning of around the spindle observed in control zygotes, mitochondria in Drp1-depleted zygotes lost their uniform distribution (*Figure 3—figure supplement 1A and B*). The biased inheritance of mitochondria was further validated by the quantification of mtDNA copy number in isolated blastomeres from 2 cell embryos (*Figure 3C*). The inheritance ratio of mtDNA in daughter blastomeres of control embryos was comparable, whereas Drp1 depletion led to biased mtDNA inheritance. Although Drp1 depletion did not affect intracellular ATP levels in the whole embryo (*Figure 2H*), we tested if the biased inheritance of mitochondria led to differences in blastomere ATP levels. Using AT1.03, ATP concentration and mitochondrial distribution were simultaneously visualized in blastomeres of 2 cell embryos (*Figure 3—figure supplement 1C*). Notably, the bias in ATP levels between blastomeres of 2 cell embryos was greater following the Drp1 depletion (*Figure 3D*). We then quantified mtDNA copy number in each blastomere and found a positive correlation between mtDNA inheritance and ATP levels in the individual blastomeres (*Figure 3E*).

The presence of ER-mitochondria MCSs may lead to mitochondria-associated ER asymmetry during cleavage. Live imaging of mitochondria and ER showed that part of the ER is confined to mitochondrial aggregation, resulting in asymmetric distribution at the telophase of the first cleavage (*Figure 3F*; *Figure 3—figure supplement 1D*; *Figure 3—video 2*), which was closely correlated with

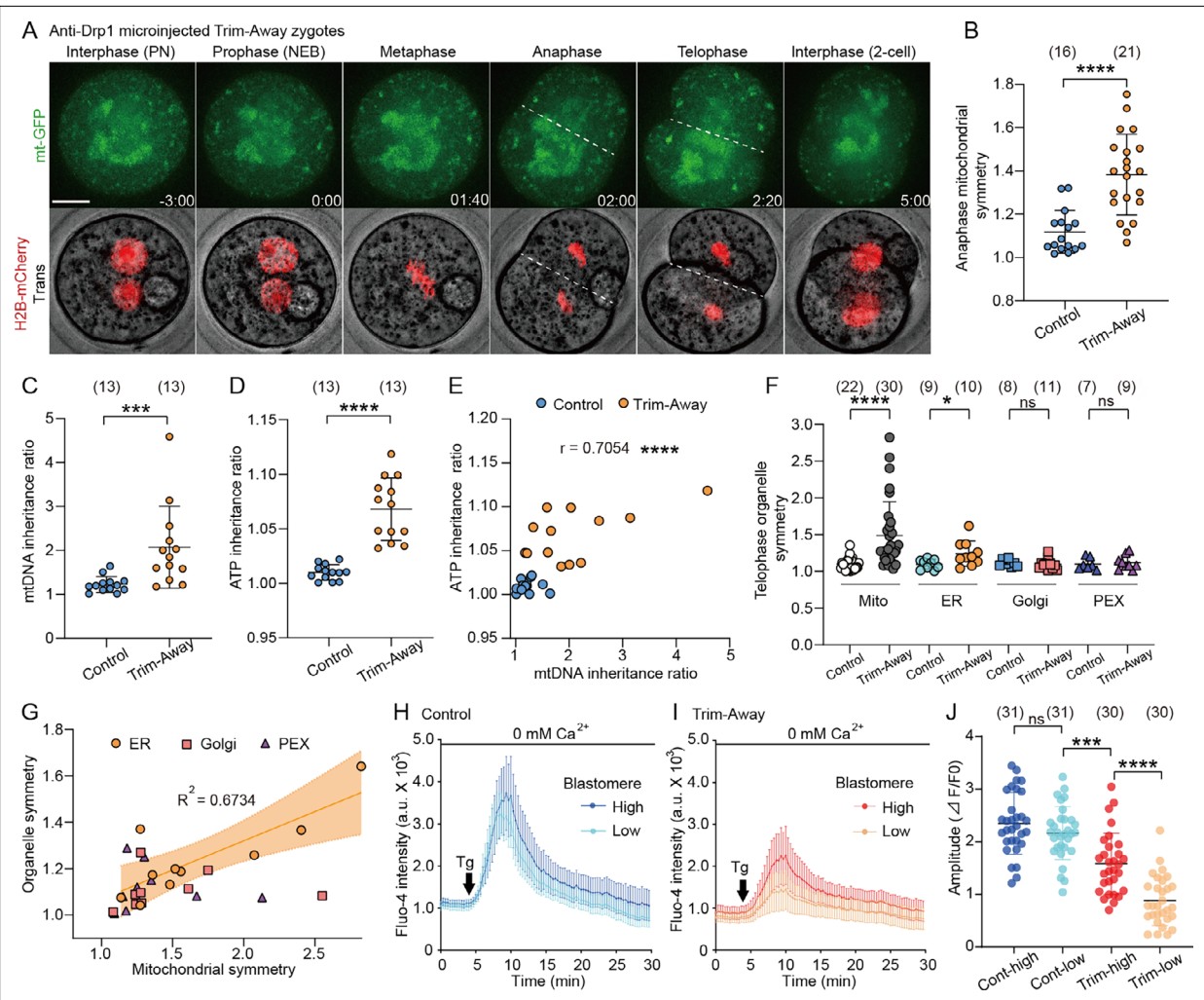

**Figure 3.** Drp1 depletion causes asymmetric mitochondrial inheritance and functions. (**A**) Representative time-lapse images (maximum-intensity Z projection) of mitochondria and chromosomes in anti-Drp1 microinjected Trim-Away zygotes expressing mt-GFP and H2B-mCherry. Time is relative to the onset of NEBD. Dashed lines indicate cleavage plane. Scale bar, 20 μm. (**B**) Quantitation of mitochondrial asymmetry at anaphase in control and Drp1-depleted zygotes. Total mitochondrial fluorescence in each dividing blastomere was measured, and the greater value was divided by the smaller value for the symmetry index. (**C**) mtDNA inheritance ratio after the first cleavage division of control and Drp1-depleted zygotes. mtDNA copy number in each daughter blastomere of 2 cell embryo was measured, and the ratio was calculated by dividing the greater value by the smaller value. (**D**) ATP inheritance ratio between two blastomeres was estimated from comparison of emission ratio of AT1.03 (YFP/CFP) in the daughter blastomere A/B, where A is the daughter blastomere with more mitochondria as confirmed by MitoTracker staining (see *Figure 3—figure supplement 1C*). (**E**) Correlation between mtDNA and ATP inheritance ratio between 2 cell blastomeres. (**F**) Quantitation of organelle asymmetry at telophase was analyzed from time-lapse images of embryos co-expressing mt-GFP with mCherry-tagged ER, Golgi, or peroxisome. Total mitochondrial fluorescence in each blastomere was measured, and the greater value was divided by the smaller value. (**G**) Organelle symmetry ratios, as shown in (**F**), are plotted against the mitochondrial symmetry ratio for each Drp1-depleted embryo. Note that mitochondrial and endoplasmic reticulum (ER) inheritance ratios are highly correlated ($R^2=0.6734$). (**H–J**) Endoplasmic reticulum $Ca^{2+}$ content in each blastomere of control (**H**) and Drp1-depleted (**I**) embryos was estimated following the addition of 10 μM thapsigargin in $Ca^{2+}$-free medium, and comparisons of the Fluo-4 fluorescent peaks are shown in a graph (**J**). Data are represented as mean ± SD and p-values calculated using unpaired two-tailed t-tests with Welch's correction (**B, C, D, and F**) or one-way ANOVA with Turkey's multiple comparison test (**J**). *$p<0.05$, ***$p<0.001$, ****$p<0.0001$; ns, not significant. Number of zygotes/embryos are indicated in brackets.

The online version of this article includes the following video and figure supplement(s) for figure 3:

**Figure supplement 1.** Drp1 depletion increases asymmetry in mitochondrial inheritance between blastomeres.

**Figure 3—video 1.** Asymmetric mitochondrial partitioning during the first cleavage of Drp1-depleted zygotes, related to *Figure 3*.
https://elifesciences.org/articles/99936/figures#fig3video1

**Figure 3—video 2.** Asymmetric partitioning of the endoplasmic reticulum (ER) confined to mitochondrial aggregation regions in Drp1-depleted

*Figure 3 continued on next page*

mitochondrial asymmetry (**Figure 3G**). No significant asymmetry was observed in the distribution of Golgi or peroxisomes, suggesting that these organelles are not directly confined by mitochondrial location. The ER is the main reservoir of intracellular $Ca^{2+}$ stores in non-excitable cells. ER-mitochondria MCSs create localized domains of high $Ca^{2+}$ concentration required for $Ca^{2+}$ transfer from the ER to the mitochondria, in which the propagation of $Ca^{2+}$ signals in the mitochondria controls energy metabolism (**Rizzuto et al., 2012**). To explore whether alterations in the ER and mitochondrial distribution and the resulting segregation of the inter-organelle contacts affect intracellular $Ca^{2+}$ homeostasis in 2 cell embryos, the $Ca^{2+}$ response in each blastomere after exposure to thapsigargin (Tg) in $Ca^{2+}$ free medium, a specific inhibitor of sarco/ER $Ca^{2+}$-ATPase, was compared between control and Drp1 Trim-Away embryos (**Figure 3H and I**). Drp1 depletion significantly reduced thapsigargin-induced $Ca^{2+}$-release and also caused significant variability between blastomeres of individual 2 cell embryos, which was not seen in controls (**Figure 3J**). These findings indicate that Drp1 depletion disrupts mitochondrial-ER interaction, leading to compromised ER $Ca^{2+}$ stores, and that asymmetric inheritance of ER leads to further variability in $Ca^{2+}$ stores between daughter blastomeres.

## Drp1 is required for the developmental competence of preimplantation embryos

We found that Drp1-depleted zygotes were predominantly arrested at the 2 cell stage and few embryos developed to the blastocyst stage (**Figure 4A and B**). This is a specific effect of Drp1 deletion because none of the internal control conditions increased arrest at the 2 cell stage and arrest was completely reversed by microinjecting Trim-away insensitive exogenous mCh-Drp1 mRNA (**Figure 4C**). Furthermore, western blot analysis confirmed expression of mCh-Drp1, in conditions where endogenous Drp1 was degraded by Trim-Away (**Figure 4D**).

More than half of the Drp1 Trim-Away embryos were arrested at the 2 cell stage, the vast majority of which did not enter the following M-phase, as assessed by the presence of the nuclear envelope (data not shown). Phosphorylated histone H3 (Ser10), localized in the heterochromatin region in G2-phase cells (**Hendzel et al., 1997**), was detected in 89% (16/18) of arrested embryos following Drp1 depletion (**Figure 4—figure supplement 1A**), suggesting that the majority of interphase arrests occur in the G2-phase. Since G2/M interphase arrest presumably activates a DNA damage checkpoint, we estimated DNA lesions by immunofluorescence of γH2AX, a widely used marker for DNA, damage 46 hr post hCG (corresponding to the G2-stage of the 2 cell stage). As expected, the number of γH2AX foci in Drp1-depleted embryos was significantly greater than that in control embryos (**Figure 4—figure supplement 1B and C**). The accumulation of reactive oxygen species (ROS), which are generated as byproducts of mitochondrial energy production, increases susceptibility to DNA damage. Intracellular ROS levels, as estimated by H2DCFDA fluorescence, were highly accumulated in mitochondria (**Figure 4—figure supplement 1D and E**). This is a specific effect of Drp1 deletion because none of the internal control conditions increased arrest at the 2 cell stage and arrest was completely reversed by microinjecting Trim-away insensitive exogenous mCh-Drp1 mRNA.

## Misplaced mitochondria during the first cleavage division impair the assembly of parental chromosomes, leading to binuclear formation

We observed that approximately 30% (17/53) of Drp1 Trim-Away embryos consisted of 3 or 4 blastomeres despite the first cleavage (indicated by arrowheads in **Figure 4B**), suggesting that cleavage defects have occurred. Live imaging experiments showed that Drp1 Trim-Away embryos frequently display chromosome segregation defects. As a typical example, two pronuclei frequently fail to reach apposition but continue into mitosis and subsequently enter anaphase, leading to binucleate blastomeres in the subsequent 2 cell embryo (**Figure 5A**; **Figure 5—video 1**). Immunostaining of metaphase zygotes showed that 55% (17/31) of Drp1-depleted zygotes formed two independent spindles with clustered microtubule organizing centers (**Figure 5B**). Both confocal and EM images showed that large mitochondrial clumps occupied the central region in Drp1-depleted zygotes, to which two

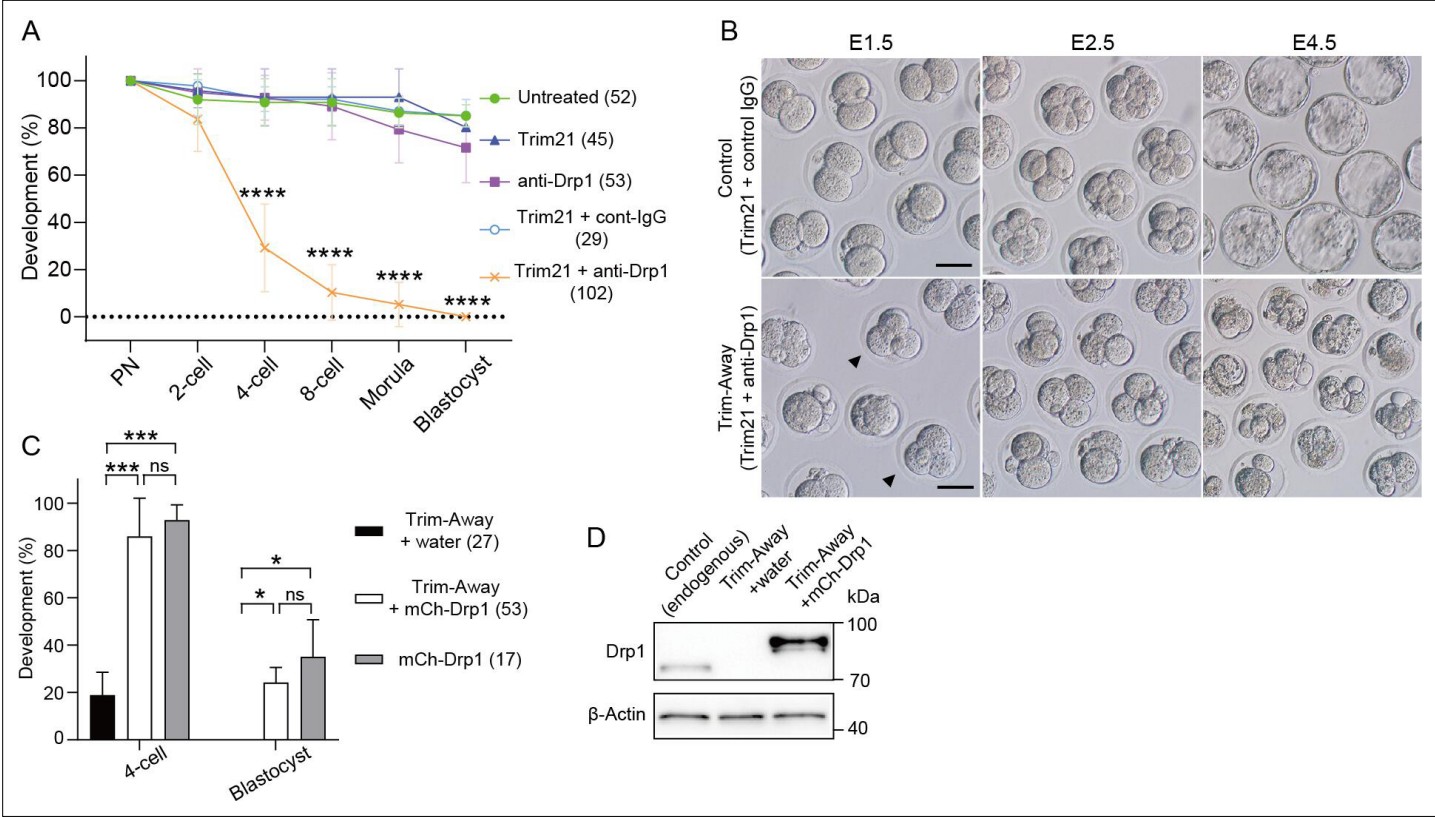

**Figure 4.** Drp1 depletion causes early embryonic arrest. (**A**) Developmental competence of control and Drp1 Trim-Away zygotes. Following Drp1 depletion by Trim-Away (co-injection of Trim21 mRNA and anti-Drp1 antibody) or the control experiments in *Figure 2B*, zygotes were cultured to the indicated stages. (**B**) Representative images of embryos microinjected with Trim21 mRNA and control IgG or anti-Drp1 were cultured. Some embryos at E1.5 have three or four blastomeres (arrowheads). Scale bar, 20 μm. (**C**) Developmental competence of Drp1-depleted embryos is rescued by microinjection of mCherry-Drp1 (mCh-Drp1) mRNA. Percentage of zygotes reaching the indicated developmental stage. Note that the 2 cell stage arrest in control condition with RNA buffer (water) upon Trim-Away depletion was reversed by microinjecting mCh-Drp1 mRNA. (**D**) Western blot analysis of control and Drp1 Trim-Away zygotes that were microinjected with water or mCh-Drp1 mRNA upon Trim-Away depletion. Lysates of 20 zygotes were extracted 5 hr after microinjection and were probed with antibodies specific to Drp1 and β-actin. Data are represented as mean ± SD and p-values calculated using one-way ANOVA with Turkey's multiple comparison test. *p<0.05, ***p<0.001, ****p<0.0001; ns, not significant. Number of embryos are indicated in brackets.

The online version of this article includes the following source data and figure supplement(s) for figure 4:

**Source data 1.** Original files of the full raw uncropped blots displayed in *Figure 4D*.

**Source data 2.** Figures with the uncropped blots with the relevant bands and treatments for *Figure 4D*.

**Figure supplement 1.** Drp1 depletion increases ROS in mitochondria and DNA damage.

spindles were separately located (*Figure 5—figure supplement 1A and B*). Tracking of two sets of chromosomes with mitochondria following Drp1 depletion revealed that the mitochondrial aggregation caused by Drp1 depletion sits between the pronuclei and subsequent spindles, such that the distance from chromosomes to the centre of the zygote is increased (*Figure 5C and D*). The duration from mitotic entry to chromosome segregation at anaphase was not affected (not shown) but binucleation was observed in 74% (32/43) of Drp1-depleted embryos and in none of the control zygotes (n=51) (*Figure 5E*). In addition to the binucleation of blastomeres via dual spindle formation, other cell division abnormalities were apparent. In approximately half (24/43) of the zygotes, multiple cleavage furrows formed between the different sets of chromosomes, leading to the transient formation of three or four blastomeres, of which 42% (10/24) failed to complete cytokinesis and collapsed back to form binucleated 2 cell embryos (*Figure 5—figure supplement 1C and D*; *Figure 5—video 2*). Together, these results suggest that proper control of mitochondrial positioning during mitosis is crucial for assembly into a single spindle, and marked mitochondrial centration causes chromosome segregation defects, leading to binuclear formation.

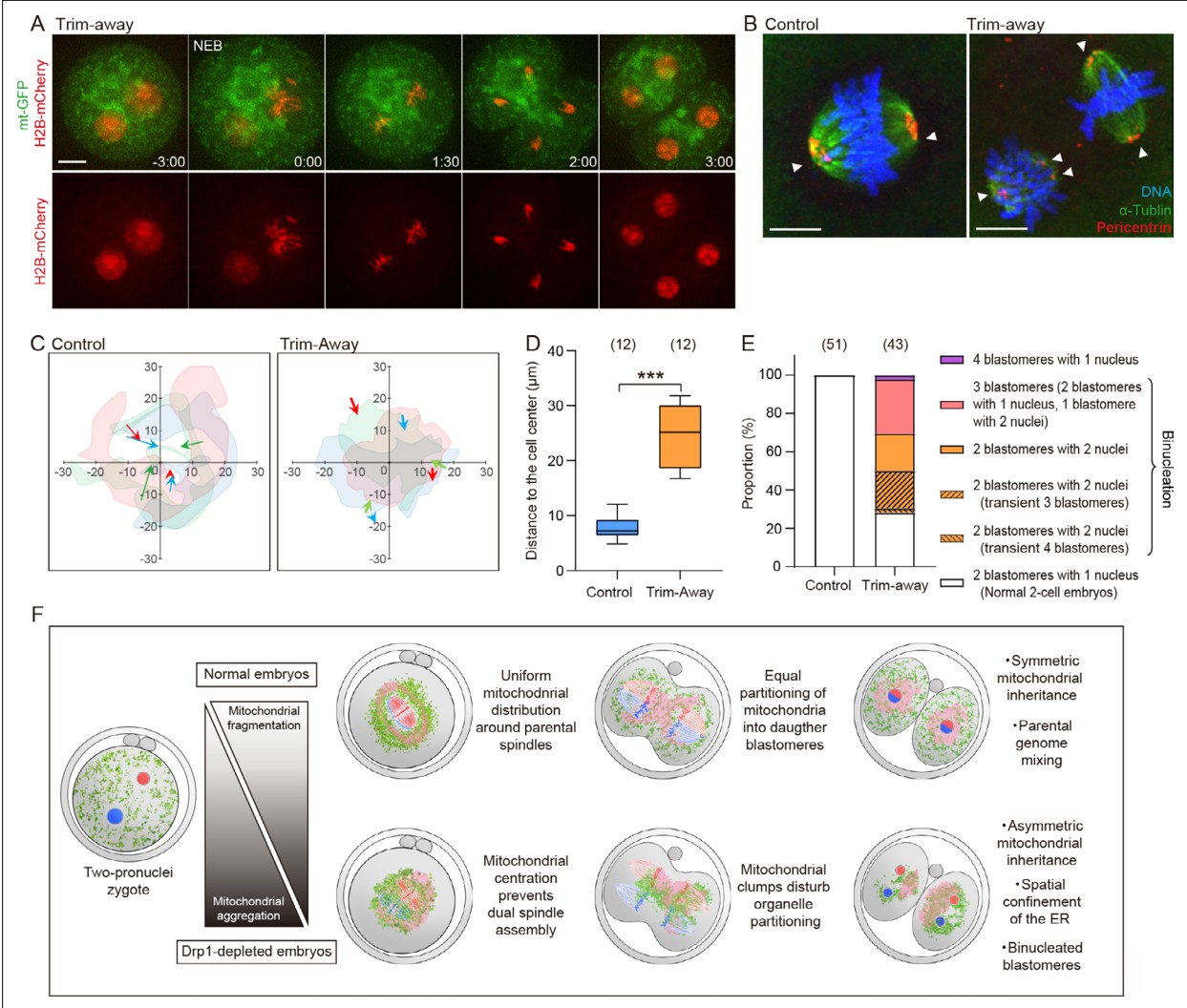

**Figure 5.** Misplaced mitochondria cause chromosome segregation defects leading to binucleation. (**A**) Representative time-lapse images (maximum-intensity Z projection) of mitochondria and chromosomes in Drp1 Trim-Away zygotes expressing mt-GFP and H2B-mCherry. Time is relative to the onset of nuclear envelope breakdown (NEB). Scale bar, 20 μm. (**B**) Representative immunofluorescence images of control (n=10) and Drp1-depleted (n=31) zygotes at metaphase of the first cleavage. Pericentrin (red), microtubules (green), and DNA (blue) were imaged by confocal microscopy. White arrowheads indicate spindle poles. Scale bars, 10 μm. (**C**) The position of chromosomes relative to the center of zygotes. The centroids of two sets of chromosomes were tracked, and only the displacements projected in the direction pointing from initial (NEB) to the final position (just before anaphase onset) were indicated. Data for three representative zygotes were presented, and each color represents a zygote. Images of mitochondria at the final position automatically had their thresholds stipulated using Fiji (Moments algorithm) and merged with images filled inside the outline with the color corresponding to the chromosomes. (**D**) Quantification of the distance of chromosomes to cell center just before anaphase onset. (**E**) Cell division abnormalities at the first cleavage. Note that binucleated blastomeres in Drp1-depleted embryos result from various chromosome segregation defects (see also *Figure 5—figure supplement 1C and D*). (**F**) Schematic summary of results highlighting significant findings in this study. Drp1 mediates mitochondrial fragmentation, ensuring symmetric mitochondrial inheritance between daughter blastomeres. In normal embryos, fragmented mitochondria (green) and the endoplasmic reticulum (ER) (pink) uniformly surround two spindles, allowing parental spindles (blue and red) to assemble at the center of metaphase zygotes. Drp1 depletion increases mitochondrial aggregation and asymmetric distribution. Misplaced mitochondria disturb the organelle positioning during the first cleavage division, leading to asymmetric organelle inheritance and chromosome missegregation, which in turn results in early embryonic arrest. Data are represented as mean ± SD and p-values calculated using unpaired two-tailed t-tests with Welch's correction. ***p<0.001.

The online version of this article includes the following video and figure supplement(s) for figure 5:

**Figure supplement 1.** Marked mitochondrial centration causes chromosome segregation defects.

**Figure 5—video 1.** Binuclear formation in Drp1-depleted embryos, related to *Figure 5*.

https://elifesciences.org/articles/99936/figures#fig5video1

*Figure 5 continued on next page*

*Figure 5 continued*

**Figure 5—video 2.** Examples of two typical cleavage defects leading to binucleation in Drp1 Trim-Away zygotes, related to *Figure 5—figure supplement 1*.

https://elifesciences.org/articles/99936/figures#fig5video2

## Discussion

### Mechanisms underlying symmetric mitochondrial inheritance in preimplantation embryos

To ensure proper partitioning of intracellular organelles for two functional daughter cells, cells undergo complex and coordinated remodeling of their cytoskeleton and membranes during cell division through mechanisms that are less well understood than chromosomal dynamics. As mitochondria are unique organelles with their own genomes, their biased inheritance is expected to have detrimental effects on daughter cell function. We showed that Drp1 mediates mitochondrial fragmentation and dispersion, enabling symmetric partitioning between daughter blastomeres stochastically, which is indispensable for healthy embryo development (*Figure 5F*).

Mitochondria that form a morphologically complex network and interact with microtubules at the interphase are rapidly fragmented upon mitotic entry, detached from the microtubules, and redistributed throughout the cytoplasm (*Ball and Singer, 1982*). Mitochondrial fission is activated via the phosphorylation of Drp1 during mitosis. Two mitotic kinases, CDK1 and Aurora A, activate Drp1, leading to enhanced mitochondrial fission (*Yamano and Youle, 2011*). As the cell exits mitosis, Drp1 is sumoylated and targeted for degradation to prevent mitochondrial fragmentation (*Figueroa-Romero et al., 2009*). Although it remains to be investigated whether these cell cycle-dependent modifications of Drp1 also occur in early embryos, mitochondria appear to be constitutively fragmented throughout the cell cycle. This unique mitochondrial morphology not only facilitates symmetric partitioning in cell division without cell growth, but also appears to be well-adapted to the rapid proliferation of single-cell zygotes into multicellular organisms before cell lineage specification. In asymmetrically dividing cells, functionally distinct mitochondria are unequally apportioned to influence daughter cell fate (*Katajisto et al., 2015*). In contrast, symmetric cell division is intended to produce identical daughter cells with comparable fates. In this context, the highly fragmented mitochondrial morphology in early embryos may contribute to avoiding cell fate divergence, as well as fulfill metabolic homogeneity among blastomeres. Interestingly, the apparent morphological changes in mitochondria occur at the blastocyst stage, as elongated mitochondria are formed in trophectoderm (TE) cells, which is likely to be associated with altered energy metabolism in the TE lineage (*Kumar et al., 2018*). Considering that the transformation from eggs to embryos is a fundamental process in animal development involving changes such as the transition from meiotic to mitotic spindles and maternal to zygotic gene expression, changes in mitochondrial morphology and function may also play a role in regulating cell fate and function in the early embryo.

The cytoskeleton is a critical regulator of organelle positioning. In contrast to a uniformly distributed mitochondria throughout the cytoplasm during somatic cell division (*Chung et al., 2016*; *Moore et al., 2021*), mitochondria progressively accumulated around pronuclei and encircled the metaphase spindle of the first embryonic cleavage. Since the two pronuclei migrate from the periphery to the centre of the eggs by the F-actin-dependent mechanism to unite chromosomes on the first mitotic spindle (*Chaigne et al., 2016*; *Scheffler et al., 2021*), actin filaments may drive the mitochondrial movement. In fact, mitochondrial distribution was closely associated with cytoplasmic F-actin organization in zygotes, especially at the metaphase. Although depolymerization of F-actin prior to metaphase led to asymmetric mitochondrial partitioning, the cell division itself was also asymmetric, probably due to the failure of centering of the pronuclei. Given that the depletion of Myo19 did not affect the mitochondrial inheritance between daughter blastomeres, the actin-based motor may not contribute the symmetric partitioning of mitochondria in cleavage embryos. This result is mostly consistent with previous observations in mitochondrial Rho GTPase (Miro) 1-deleted zygotes. Miro1 is localized in the mitochondrial outer membrane and the adaptor protein TRAK mediates the attachment of motor proteins to the mitochondria (*López-Doménech et al., 2018*). Miro also regulates Myo19, ensuring stability to actin-mediated mitochondrial distribution (*López-Doménech et al., 2018*), and thus loss of Miro leads to asymmetric segregation of mitochondria during mitosis, most likely due to reduced

levels of Myo19. In Miro1-deleted zygotes, mitochondrial dynamics was partially disturbed, but this did not affect mitochondrial partitioning to daughter blastomeres, and fertility was normal as a result (*Lee et al., 2022*). Thus, fragmented mitochondria may be primarily important for symmetric mitochondrial partitioning in cleavage embryos, as a less precise stochastic partitioning mechanism may be suitable or sufficient for equally partitioning of mitochondria, which are present in abundance and are located across the cell volume in oocytes/zygotes. Nevertheless, it is also worth considering the possibility that actin accumulated in mitochondria may be indirectly involved in mitochondrial dynamics via mitochondrial fission. For example, inverted formin 2 controls actin polymerization and has been reported to be required for efficient mitochondrial fission as an upstream function of Drp1 (*Korobova et al., 2013*).

## The physiological roles of mitochondrial fission in preimplantation embryos

Inhibition of Drp1-mediated mitochondrial fission causes severe embryonic arrest. Given the housekeeping cellular functions of mitochondria, dysregulation of mitochondrial dynamics may exert profound effects on the developmental competence of embryos. Mitochondrial dynamics are a highly regulated process, and when disrupted, can lead to cellular and tissue pathologies. The importance of mitochondrial fission has been intensively studied in animal cells, and emerging evidence suggests that Drp1-mediated fission participates in diverse cellular processes, including ER contacts, $Ca^{2+}$ signaling, autophagy, and apoptosis (*Kraus et al., 2021*). Drp1-KO mice are embryonic lethal, most likely due to brain, heart, and placental defects; however, various tissues appear normal, and cultured cell lines continue to proliferate even though mitochondria are unevenly segregated (*Ishihara et al., 2009*; *Wakabayashi et al., 2009*). Cells lacking Drp1 have highly elongated mitochondria that cannot be divided into transportable units, and typically accumulate in a limited area, leaving large parts of the cell devoid of mitochondria. This imbalanced distribution of mitochondria may fail to meet local energy demands and thus would be more manifest in large or extended cells, such as neurons and oocytes/zygotes. Moreover, our data demonstrated that Drp1 depletion in cleavage embryos caused biased mitochondrial inheritance and energy heterogeneity between blastomeres. Given the low levels of de novo mtDNA replication during preimplantation development, these biases may be further magnified by cleavage division progression. It is also noteworthy that we showed that mitochondrial aggregation induced by the loss of Drp1 disturbs the spatial organization of the ER. The aggregated ER subdomains likely caused leakage of $Ca^{2+}$ stores, which may compromise the tethering functions at the ER MCSs. The precise mechanisms of 2 cell stage arrest arising from depletion of Drp1 remain elusive. It has been reported that Drp1 depletion during meiosis leads to premature chromosome segregation due to spindle assembly checkpoint defects (*Zhou et al., 2022*). Drp1-deficient cells has also been reported to cause replication stress, leading to DNA damage response and the G2/M cell cycle arrest (*Qian et al., 2012*), which is more consistent with our observations. The presence of a Chk1-dependent checkpoint leading to G2 arrest in 2 cell mouse embryos (*Ladstätter and Tachibana-Konwalski, 2016*) suggests that this surveillance mechanism may be involved in the 2 cell arrest process in this study.

In the present study, we found frequent binucleation in Drp1-depleted embryos. Binucleation of early embryos is relatively common in human IVF procedures, but the mechanisms of its occurrence and association with the developmental competence of the embryo are not fully understood (*Gomes Paim and FitzHarris, 2020*). *Reichmann et al., 2018* recently demonstrated that mouse zygotes form two functionally independent spindles before anaphase, and spatially separate parental genomes during the first cleavage. Notably, failure to align the two spindles produced errors, resulting in binuclear formation in the 2 cell embryos. The present study demonstrates that clustered mitochondria in the central space of zygotes interfere with the assembly of two spindles, implying a potential contribution of the organelle misplacement to binucleation in human embryos.

## Mitochondrial dynamics and quality control

The present study revealed the response to defective mitochondrial fission at the cellular level in early embryos, but the response at the organelle level remains to be elucidated. Mitochondria in oocytes, with hundreds of thousands of mtDNA copies, are maternally inherited after fertilization. The preservation and apportionment of healthy mtDNA in oocytes and early embryos is important

for the inheritance of normal mtDNA in the offspring. Recent full-length mtDNA sequencing has identified de novo mtDNA mutations in oocytes, and their frequency increases with age, implying mtDNA turnover during meiotic arrest (*Abrisch et al., 2020*). Oocytes with severe mtDNA mutations can be eliminated (*Fan et al., 2008*), although the thresholds at which mtDNA mutations and the resulting mitochondrial dysfunction result in selection remain unclear. The autophagic elimination of mitochondria and mitophagy may also function as filters to segregate and eliminate pathogenic mtDNA mutations at the organelle level (Youle and *Youle and van der Bliek, 2012*). An attractive model has been proposed for the *Drosophila* female germline, as fragmentation of the mitochondrial network facilitates mitophagy and allows selective elimination of defective mitochondria containing mutant mtDNA (*Lieber et al., 2019*). Functional links between mitochondrial fission and the removal of damaged mitochondria by mitophagy have also been documented in mammalian cells, as Drp1 deficiency causes mitochondrial dysfunction owing to the failure of a Drp1-dependent mechanism of mitophagy that removes damaged mitochondria within the cell (*Twig et al., 2008*). The highly fragmented mitochondrial structure in mammalian oocytes/zygotes allows mtDNA to be sequestered in small units, which may be beneficial for eliminating mutation-impaired mitochondria for mitophagy. Furthermore, symmetric segregation of fragmented mitochondria might randomize mutant mtDNA among daughter blastomeres during embryonic cleavage to maintain tissue homeostasis. Elucidating the behavior of mutant mitochondria in early embryos is clinically meaningful. Mitochondrial replacement therapy (MRT) is a promising approach to prevent the transmission of mtDNA disorders from mother to child, as pathogenic mutant mtDNA in mature oocytes or zygotes is replaced with normal mtDNA by nuclear transfer (*Hyslop et al., 2016*; *Kang et al., 2016*). A potential problem with MRT is that a small fraction of patients' mtDNA is inevitably carried over into the reconstructed oocytes or zygotes (approximately 1 to 4%, respectively). It has been noted that these low levels of mutant mtDNA may predominantly expand during post-implantation embryogenesis, suggesting a rapid shift to one mtDNA haplotype during early development (*Lee et al., 2012*). Therefore, it is important to investigate the role of mitochondrial dynamics in the regulation of mtDNA mutations in the preimplantation embryo, and whether mitochondrial fission contributes to reducing the mutational load.

# Materials and methods

### Key resources table

| Reagent type (species) or resource | Designation | Source or reference | Identifiers | Additional information |
|---|---|---|---|---|
| Genetic reagent (*Mus musculus*) | DBA/2 (♂) | Charles River Laboratories | RRID:MGI:2159768 | |
| Genetic reagent (*M. musculus*) | Mouse: C57BL/6 N (♀) | Charles River Laboratories | RRID:MGI:2159965 | |
| Genetic reagent (*M. musculus*) | Mouse: B6D2F1 mice | Department of Animal Resources, Okayama University | RRID:MGI:5649818 | |
| Genetic reagent (*M. musculus*) | $Drp1^{fl/fl}$ mice | PMID:19752021 | | |
| Genetic reagent (*M. musculus*) | $Drp1cKO$ mice | PMID:35704569 | | |
| Antibody | Mouse monoclonal anti-Drp1 | BD Biosciences | Cat# 611739; RRID:AB_399215 | WB(1:1000) Trim-Away experiment (1000 ng/mL) |
| Antibody | Mouse monoclonal anti-β-actin | Cell Signalling Technology | Cat# 3700; RRID:AB_2242334 | WB(1:2000) |
| Antibody | Rabbit polyclonal anti-Myo19 | Abcam | Cat# ab174286; RRID:AB_3712164 | IF(1:300) WB(1:1000) |
| Antibody | Rabbit polyclonal anti-Pericentrin | Abcam | Cat# ab4448; RRID:AB_304461 | IF(1:500) |
| Antibody | Mouse monoclonal Anti-γH2A.X | Abcam | Cat# ab22551; RRID:AB_447150 | IF(1:500) |

*Continued on next page*

*Continued*

| Reagent type (species) or resource | Designation | Source or reference | Identifiers | Additional information |
|---|---|---|---|---|
| Antibody | HRP anti-Mouse | Cell Signaling Technology | Cat# 7076; RRID:AB_330924 | WB(1:2000) |
| Antibody | HRP anti-Rabbit | Cell Signaling Technology | Cat# 7074; RRID:AB_2099233 | WB(1:2000) |
| Antibody | Alexa Fluor 488 anti-Mouse IgG | Thermo Fisher Scientific | Cat# A-11001; RRID:AB_2534069 | IF(1:300) |
| Antibody | Alexa Fluor 568 anti-Rabbit IgG | Thermo Fisher Scientific | Cat# A-11011; RRID:AB_143157 | IF(1:500) |
| Antibody | Normal mouse IgG | Santa Cruz Biotechnology | Cat# sc-2025; RRID:AB_737182 | Trim-Away experiment (1000 ng/mL) |
| Antibody | Normal Rabbit IgG | Cell Signaling Technology | Cat# 2729; RRID:AB_1031062 | Trim-Away experiment (1000 ng/mL) |
| Recombinant DNA reagent | pcDNA6-Mt-GFP(plasmid) | PMID:25113836 | | |
| Sequence-based reagent | pcDNA6-Mt-DsRed (plasmid) | PMID:25113836 | | |
| Recombinant DNA reagent | pcDNA6-ER-mCherry (plasmid) | PMID:25113836 | | |
| Recombinant DNA reagent | pcDNA6-H2B-mCherry (plasmid) | PMID:25113836 | | |
| Recombinant DNA reagent | pcDNA6-EB3-GFP (plasmid) | PMID:25113836 | | |
| Recombinant DNA reagent | pcDNA3-AT1.03 (plasmid) | PMID:19720993 | | |
| Recombinant DNA reagent | pcDNA3-AT1.03 RK (plasmid) | PMID:19720993 | | |
| Recombinant DNA reagent | pCMV-Trim21 (plasmid) | ORIGENE | Cat#: MR207378 | |
| Sequence-based reagent | pcDNA6-Golgi-mCherry cloning (sense) | Eurofins | PCR primers | 5'-ATCAAGCTTGCCACCATGGGCAACTTGAAG-3' |
| Sequence-based reagent | pcDNA6-Golgi-mCherry cloning (antisense) | Eurofins | PCR primers | 5'-TATCTCGAGACCACCTCCACCTCCTCCA-3' |
| Sequence-based reagent | pcDNA6-PEX-mCherry cloning (sense) | Eurofins | PCR primers | 5'-GATCTCGAGCTCAAGCTTCGAATTCTGCAG-3' |
| Sequence-based reagent | pcDNA6-PEX-mCherry cloning (antisense) | Eurofins | PCR primers | 5'- TATCTAGAGTCGCGGCCGCTACAGCTTG-3' |
| Sequence-based reagent | pcDNA6-mCherry-Drp1 infusion cloning (sense) | Eurofins | PCR primers | 5'- CAGTGTGGTGGAATTGCCACCATGGTGAGCAAG-3' |
| Sequence-based reagent | pcDNA6-mCherry-Drp1 infusion cloning (antisense) | Eurofins | PCR primers | 5'- ACTGTGCTGGATATCTCACCAAAGATGAGTCTCC-3' |
| Sequence-based reagent | mtDNA qPCR (left) | Eurofins | PCR primers | 5'-CCTATCACCCTTGCCA-3 |
| Sequence-based reagent | mtDNA qPCR (right) | Eurofins | PCR primers | 5'-GAGGCTGTTGCTTGTG-3' |
| Commercial assay or kit | RiboMAX Large Scale RNA Production Systems | Promega | Cat# P1300 | |
| Commercial assay or kit | Ribo m7G Cap Analog | Promega | Cat# P1712 | |
| Commercial assay or kit | T7 mMessage mMachine Kit | Thermo Fisher Scientific | Cat# AM1344 | |

*Continued on next page*

*Continued*

| Reagent type (species) or resource | Designation | Source or reference | Identifiers | Additional information |
|---|---|---|---|---|
| Commercial assay or kit | Poly(A) Tailing Kit | Thermo Fisher Scientific | Cat# AM1350 | |
| Chemical compound, drug | Hyaluronidase | Sigma-Aldrich | Cat# D2650 | |
| Chemical compound, drug | Protease | Sigma-Aldrich | Cat# P5147 | |
| Chemical compound, drug | DMSO | Sigma-Aldrich | Cat# D2650 | |
| Chemical compound, drug | Cytochalasin B | Sigma-Aldrich | Cat# C6762 | |
| Chemical compound, drug | Polyvinylpyrrolidone (PVP) Solution | Irvine Scientific | Cat# 90123 | |
| Chemical compound, drug | Thapsigargin | Thermo Fisher Scientific | Cat# T9033 | |
| Chemical compound, drug | MitoTracker Red CMXRos | Thermo Fisher Scientific | Cat# M7512 | |
| Chemical compound, drug | CM-H2-DCFDA | Thermo Fisher Scientific | Cat# C6827 | |
| Chemical compound, drug | Fluo-4, AM | Thermo Fisher Scientific | Cat# F14201 | |
| Chemical compound, drug | Pluronic F-127 | Thermo Fisher Scientific | Cat# P6867 | |
| Chemical compound, drug | Triton X-100 | Sigma-Aldrich | Cat# T9284 | |
| Chemical compound, drug | DAPI | Sigma-Aldrich | Cat# D9542 | |
| Software, algorithm | Fiji | PMID:22743772 | RRID:SCR_002285 | |
| Software, algorithm | QuantEv Icy plugin | PMID:30091700 | | |
| Software, algorithm | HCImage | Hamamatsu photonics | RRID:SCR_015041 | |
| Software, algorithm | Prism 8 | GraphPad Prism | RRID:SCR_002798 | |

## Preparation of zygotes

Zygotes were collected from the oviducts of 8- to 12-week-old BDF1 female mice mated with 3- to 6-month-old BDF1 male mice. Females were superovulated by intraperitoneal injection of 7.5 IU of equine chorionic gonadotropin (PMSG; ASKA Pharmaceutical, Tokyo, Japan) followed by injection of human chorionic gonadotropin (hCG; ASKA Pharmaceutical) 48 hr later. Female mice were euthanized 20–24 hr later and fertilized zygotes were retrieved in Hepes-buffered KSOM (H-KSOM) medium under paraffin oil, at 37 °C in a humidified atmosphere containing 5% $CO_2$. All mice were housed in a pathogen-free environment in filter-top cages and fed a standard diet. Animal experiments were conducted according to the guidelines of the Animal Care and Use Committee of Okayama University.

For experiments in *Figure 3C*, Drp1$^{fl/fl}$ mice (*Wakabayashi et al., 2009*) were crossed with transgenic mice that carried *Gdf-9* promoter–mediated Cre recombinase (*Lan et al., 2004*). After multiple rounds of crossing, homozygous Drp1 conditional KO female mice lacking Drp1 in oocytes (Drp1$^{fl/fl}$; Gdf9-Cre) or Drp1$^{Δ/Δ}$ oocytes were obtained. Mice that do not carry the *Cre* transgene are referred to as Drp1$^{fl/fl}$ and were used as controls. For the collection of ovulated MII oocytes, mice were superovulated by sequential intraperitoneal injections of 8 IU PMSG and 8 IU hCG at an interval of 48 hr, and the mice were culled 14–16 hr after hCG injection to collect oocytes from the oviducts. Animal experimentation was approved by Monash University Animal Experimentation Ethics Committee and was performed in accordance with Australian National Health and Medical Research Council Guidelines on Ethics in Animal Experimentation.

## Plasmids

EGFP, mCherry, and DsRed were cloned into the pcDNA6 vector between the XhoI and XbaI sites (pcDNA6/Myc-His B; Invitrogen, Waltham, MA) with a poly(A)-tail of 84 nucleotides (*Wakai et al., 2014*). The gene sequence encoding the mitochondrial targeting sequence of Cox VIII was amplified by PCR and ligated to the EGFP and DsRed-bearing pcDNA6 (mt-GFP and mt-DsRed). The gene sequences encoding the ER-targeting sequence of calreticulin and the KDEL ER retention sequence were cloned into mCherry (ER-mCherry)-bearing pcDNA6. For visualization of DNA and microtubule growth, gene sequences encoding histones H2B and EB3 were ligated to the pcDNA6 vector in frame with mCherry (H2B-mCherry) and EGFP (EB3-GFP), respectively. The gene sequences encoding the amino-terminal 33 residues of endothelial nitric-oxide synthase as a Golgi targeting sequence (Addgene, 14873) and peroxisomal targeting signal 1 (Addgene, 54520) were cloned into mCherry-bearing pcDNA6 (Golgi-mCherry and PEX-mCherry). The gene sequences encoding N-terminally mCherry-tagged Drp1 (Addgene, 49152) were inserted between the EcoRI and PstI sites of the pCDNA vector (mCherry-Drp1). ATP biosensor, AT1.03 and AT1.03RK in the pcDNA3.1 vector were kindly provided by Dr H. Imamura (Kyoto University). Capped mRNA was synthesized with T7 polymerase (RiboMAX Large-Scale RNA Production, Promega, Madison, WI) according to the manufacturer instructions. mRNAs were delivered into zygotes using a piezo-driven micropipette unit (Prime Tech, Tsuchiura, Japan). For microinjection, mRNA solution was loaded into glass micropipettes at the concentration of 200 ng/µl (mt-GFP, EB3-GFP, mt-DsRed, H2B-mCherry, ER-mCherry, Golgi-mCherry, and PEX-mCherry), 500 ng/µl (mCherry-Drp1), or 1000 ng/µl (AT1.03 and AT1.03RK). The volumes injected typically ranged from 2 to 10 pl, which is 1–5% of the total volume of the zygotes. Manipulation was carried out in H-KSOM medium containing 5 mg/mL cytochalasin B (Sigma, St. Louis, MO) to increase post-microinjection survival.

## Trim-Away

Trim-Away was performed as previously described with some modifications (*Clift et al., 2017*). In brief, pCMV6-Entry vector harboring Trim-21 mouse ORF Clone (Origene, Rockville, MD) was linearized by FseI, and mRNA was synthesized using T7 mMessage mMachine and poly-A tailed with Poly(A) tailing kit (Thermo Fisher Scientific, Waltham, MA). Mouse monoclonal anti-Drp1 antibody (BD Biosciences, Franklin Lakes, NJ) or rabbit monoclonal anti-Myo19 antibody (Abcam, Cambridge, UK) was concentrated using Amicon Ultra-0.5 100 KDa centrifugal filter devices (Millipore, Bedford, MA), and the buffer was replaced with PBS containing 0.03% NP40 (Nakarai, Kyoto, Japan). Pronuclear stage zygotes were injected with Trim21 mRNA (1000 ng/mL) and subsequently with anti-Drp1 or anti-Myo19 antibodies. For the control experiment, normal mouse immunoglobulin G (IgG) antibody (Santa Cruz Biotechnology, Dallas, TX) and rabbit IgG (Cell Signaling Technology, Danvers, MA) was used instead of an anti-Drp1 antibody and anti-Myo19 antibodies, respectively.

## Western blotting

Whole cell lysates from zygotes/embryos were prepared by adding a 2x sample buffer. Proteins were separated by SDS-PAGE and transferred to PVDF membranes (Millipore). The membranes were then blocked and probed with anti-Drp1 (1:1000) or anti-Myo19 (1:1000) antibodies for 1 hr at room temperature. Goat anti-mouse antibody conjugated to horseradish peroxidase (HRP) was used as a secondary antibody (1:2000) for chemiluminescence detection (Clarity Western ECL Substrates, Bio-Rad, Hercules, CA) according to the manufacturer's instructions. The signal was digitally captured using a ChemiDoc XRS + imaging system (Bio-Rad). The same membranes were stripped at 50 °C for 30 min (62.5 mM Tris, 2% SDS, and 100 mM 2-beta mercaptoethanol) and re-probed with anti-β-actin monoclonal antibody (1:2000, Cell Signaling Technology) as the internal control.

## mtDNA copy number

The zona pellucida of 2 cell embryos was removed with H-KSOM medium containing 0.1% protease (Sigma, St. Louis, MO). Individual blastomeres from the zona-free embryos were obtained by repeatedly pipetting in Ca$^{2+}$-free and Ma$^{2+}$-free H-KOSM medium. After washing in PBS, each blastomere was transferred into 5 µL of lysis buffer 50 mM Tris-HCl, pH 8.5, with 0.5% Tween 20 and 100 µg proteinase K, at 55 °C, followed by heat inactivation at 95 °C for 10 min. The plasmid standard was constructed by cloning a 194 bp fragment of the NADH-ubiquinone oxidoreductase chain 1. The

standard stock was serially diluted to prepare the standard curve ($10^3$–$10^7$). Real-time fluorescence-monitored quantitative PCR using the primer pair 5′-CCTATCACCCTTGCCA-3′ and 5′-GAGGCTGT TGCTTGTG-3′ was performed on a LightCycler 96 System (Roche, Basel, Switzerland). Each reaction of 20 µL consisted of 10 µL of SYBR Green I (Roche), 0.5 µM each of the forward and reverse primers, and 2 µL of DNA template. The cycling conditions were as follows: initial denaturation at 95 °C for 5 min, followed by 45 cycles of denaturation at 95 °C for 10 sec, annealing at 50 °C for 20 s, and elongation at 72 °C for 8 s. Standard curves were created for each run, and the sample copy number was determined from the equation relating the Ct value against copy number for the corresponding standard curve.

## Immunofluorescence

Zygotes/embryos were fixed and permeabilized with 2% paraformaldehyde in phosphate-buffered saline (PBS) containing 0.1% Triton X-100 for 40 min at room temperature. After washing with PBS supplemented with 1% bovine serum albumin (PBS–BSA), the zygotes/embryos were incubated overnight at 4 °C with mouse anti-α-tubulin (1:300, Sigma), rabbit monoclonal anti-Myo19 antibody (1:300, Abcam), rabbit anti-pericentrin (1:500, Abcam) or mouse monoclonal Anti-γH2A.X (1:500, Abcam) antibodies in PBS–BSA. Then zygotes/embryos were washed with PBS–BSA and incubated with Alexa Fluor 488-conjugated goat anti-rabbit IgG (1:300) for 1 hr at room temperature. DNA was stained with DAPI. Samples were placed in PBS under mineral oil in glass-bottom dishes (Matsunami, Osaka, Japan) and examined using a laser-scanning confocal microscope (FV1200, Olympus, Tokyo, Japan) outfitted with a ×63, 1.4 NA oil immersion objective lens.

## Mitochondrial and ROS staining

Mitochondrial distribution was analyzed by staining with 1 µM MitoTracker Red CM-H2XRos (Thermo Fisher Scientific) in H-KSOM medium for 30 min at 37 °C. Intracellular ROS levels were analyzed using 5 µM H2DCFDA (Thermo Fisher Scientific) in H-KSOM medium for 15 min at 37 °C. The embryos were placed in drops of H-KSOM medium under mineral oil in glass-bottom dishes (Matsunami), and fluorescence images of MitoTracker Red CMXros and H2DCFDA were obtained using a laser-scanning confocal microscope (FV1200) fitted with a ×63, 1.4 NA oil-immersion objective lens.

## Electron microscopy

Zygotes/embryos were fixed in 2% glutaraldehyde in a 0.1 M phosphate buffer (pH 7.4) for 1 hr and stored at 4 °C until processed. After post-fixation in 2% osmium tetroxide at 4 °C, the specimens were dehydrated with a graded series of ethanol and were embedded in epoxy resin Quetol-812. Semi-thin sections were cut for light microscopy and stained with toluidine blue for further sectioning of areas. Thereafter, ultra-thin sections were cut and stained with uranyl acetate and lead citrate and examined by transmission electron microscopy (TEM) (JEM 1200EX, JEOL) at 80 kV.

## Fluorescence resonance energy transfer and Ca²⁺ imaging

ATeam, a fluorescence resonance energy transfer-based ATP indicator, has been used successfully to measure cellular ATP levels in live somatic cells (*Imamura et al., 2009*). To estimate the relative changes in ATP levels, the emission ratio of AT1.03 and AT1.03RK (YFP/CFP) was imaged using a CFP excitation filter, dichroic beam splitter, and CFP and YFP emission filters (Chroma Technology, Rockingham, VT; ET436/20 x, 89007bs, ET480/40 m, and ET535/30 m). To measure cytoplasmic $Ca^{2+}$, embryos were incubated with 1.25 µM Fluo-4 (Thermo Fisher Scientific) supplemented with 0.02% pluronic acid (Thermo Fisher Scientific) for 20 min at room temperature. Fluo-4 was excited with 480 nm wavelengths every 20 s and the emitted light was collected at wavelengths greater than 510 nm. To estimate $Ca^{2+}$ levels in the ER, $Ca^{2+}$ rise was analyzed following the addition of 10 µM thapsigargin (Sigma) in a $Ca^{2+}$-free medium. For the monitoring of ATP and $Ca^{2+}$ levels, embryos were attached to glass-bottomed dishes (Matsunami) and placed on the stage of an inverted microscope. The fluorescence images were obtained by a scientific CMOS (sCMOS) camera (Hamamatsu Photonics, Hamamatsu), and the rotation of excitation and emission filter wheels was controlled using the MAC5000 filter wheel/shutter control box (Ludl Electronic Products Ltd, Hawthorne, NY) and HCImage software.

## Live cell confocal imaging

Spinning disk images of mitochondria (mt-GFP and mt-DsRed), chromosomes (H2B-mCherry), microtubules (EB3-GFP), ER (ER-mCherry), Golgi (Golgi-mCherry), and peroxisome (PEX-mCherry), were obtained using a confocal scanner unit (CellVoyager CV1000, Yokogawa, Tokyo, Japan) equipped with a C-Apochromat ×40, 1.2 NA water immersion objective. Images were typically acquired every 10–20 mins on 10 planes covering a 35–40 µm range (every 2.5–3.3 µm). Laser wavelengths of 488 and 561 nm were used for the excitation of EGFP and mCherry (DsRed), respectively (not exceeding 0.2% laser intensity to minimize light-induced phototoxic stress).

## Image analysis

Fiji (*Schindelin et al., 2012*) was used for the quantification of fluorescence images.

For quantification of mitochondria at nuclear or spindle periphery in *Figure 1B* and *Figure 1—figure supplement 1C*, the fluorescence images of nucleus (H2B-mCherry) of spindle (EB3-GFP) were applied with a Gaussian filter (radius: 2 pixels) to remove noise and segmented by intensity thresholding. The masked binary images were dilated with 10 pixels width and the nucleus/spindle periphery was segmented by the subtraction of the original image from the dilated image. The fluorescence intensity ratio of mitochondrial layers at the periphery to that of outside the periphery was measured to estimate mitochondrial accumulation around the nucleus/spindle.

For quantitation of mitochondrial symmetry at the anaphase in *Figure 1—figure supplement 2B* and *Figure 3B*, the cleavage plane of the dividing blastomere was manually outlined with respect to the middle of the anaphase chromosomes. Total mt-GFP intensity in each blastomere was measured, and the greater value was divided by the smaller value for the symmetry index. To analyze organelle symmetry at the telophase in *Figure 3F*, total pixel intensity in the channel corresponding to mt-GFP, ER-mCherry, Golgi-mCherry, and PEX-mCherry in each blastomere was measured. The greater value was divided by the smaller value; a ratio of 1 indicates symmetry, whereas higher values indicate asymmetry.

To measure the size and count the number of mitochondrial clusters in *Figure 2G, H*, and *Figure 2—figure supplement 2B, C*, the fluorescence images of mitochondria (mt-GFP) were analyzed using a basic measurement function (analyze particles, size) after adjusting intensity threshold (Yen algorism). Based on the average size of mitochondria in EM images, mt-GFP signals larger than 0.15 µm$^2$ were counted as mitochondrial clusters.

Uniformity analysis of mitochondrial distribution around spindles in y in *Figure 3—figure supplement 1B* was performed based on the angular distributions of mitochondria (mt-DsRed) and microtubules (EB3-GFP) using the 'QuantEv Icy' plugin (*Pécot et al., 2018*). The fluorescence images were applied with a Gaussian filter (radius: 2 pixels) to reduce noise, and xy coordinates were converted to polar coordinates. The fluorescence intensity profiles for mt-DsRed relative to the centroid of the EB3-GFP signals were divided into sectors, each 60 degrees based on the long axis of the spindle. The standard deviation of averaged mitochondrial intensity per 60° sector were calculated; higher values were considered to indicate increased non-uniformity.

For quantitation of ROS enrichment in mitochondria in *Figure 4—figure supplement 1E*, the fluorescence images of mitochondria (MitoTracker Red CMXros) were background-subtracted (rolling ball radius: 5 pixels) and applied with the Gaussian filter (radius: 2 pixels) to remove noise. The inside mitochondria was masked by intensity thresholding (Moments algorism). The outside mitochondria was segmented by subtracting the masked image from the whole embryo area. The intensity ratio of H2DCFDA fluorescence inside mitochondria to outside mitochondria was measured to estimate ROS accumulation in mitochondria.

For tracking nuclei in zygotes in *Figure 5C and D*, XY coordinates of the centroid of male and female chromosomes (H2B-mcherry) relative to the center of the zygote were obtained using the 'Manual tracking' plugin. Only the zygotes with nuclei that remained in the focal plane were analyzed.

## Statistical analysis

All statistical analyses were performed using GraphPad Prism software. Values from three or more experiments performed on different batches of cells were analyzed by statistical tests and p-values are indicated in figures and figure legends.

## Resource availability

All unique/stable reagents and plasmids generated in this study are available from the lead contact with a completed Materials Transfer Agreement. Further information and requests for resources and reagents should be directed to and will be fulfilled by the lead contact, Takuya Wakai (t2wakai@okayama-u.ac.jp).

## Acknowledgements

We thank Dr. H Imamura (Kyoto University) for sharing AT1.03 and AT1.03RK plasmids. We are thankful to Dr. Hiromi Sesaki (Johns Hopkins University School of Medicine) for sharing *Drp1*<sup>fl/fl</sup> mice line. We thank Ms. Sayaka Nakato and the staff of Mio Fertility Clinic for technical assistance with CellVoyager CV1000. This work has been partially supported by the Core-Facility Portal (CFPOU) at Okayama University (DGP_13).

## Additional information

### Funding

| Funder | Grant reference number | Author |
|---|---|---|
| Japan Society for the Promotion of Science | 17K17905 | Takuya Wakai |
| National Health and Medical Research Council | 1165627 | John Carroll Deepak Adhikari |
| Japan Society for the Promotion of Science | 20K06627 | Takuya Wakai |
| Japan Society for the Promotion of Science | 23K0886905 | Takuya Wakai |

The funders had no role in study design, data collection and interpretation, or the decision to submit the work for publication.

### Author contributions

Haruna Gekko, Ruri Nomura, Formal analysis, Investigation; Daiki Kuzuhara, Data curation, Investigation; Masato Kaneyasu, Data curation, Visualization; Genpei Koseki, Data curation, Formal analysis; Deepak Adhikari, Formal analysis, Funding acquisition, Investigation; Yasuyuki Mio, Resources; John Carroll, Supervision, Funding acquisition, Writing – review and editing; Tomohiro Kono, Supervision; Hiroaki Funahashi, Resources, Supervision; Takuya Wakai, Conceptualization, Data curation, Formal analysis, Supervision, Validation, Investigation, Writing – original draft

### Author ORCIDs

Haruna Gekko https://orcid.org/0009-0006-3847-4967
Ruri Nomura https://orcid.org/0009-0005-0003-5179
Daiki Kuzuhara https://orcid.org/0009-0000-7941-4457
Masato Kaneyasu https://orcid.org/0009-0008-2442-0444
Genpei Koseki https://orcid.org/0009-0007-6074-9813
Deepak Adhikari https://orcid.org/0000-0002-3404-8753
Yasuyuki Mio https://orcid.org/0000-0002-0413-6728
John Carroll https://orcid.org/0000-0001-9644-5861
Tomohiro Kono https://orcid.org/0009-0006-9456-916X
Hiroaki Funahashi https://orcid.org/0000-0002-9683-328X
Takuya Wakai https://orcid.org/0000-0003-4705-8974

### Ethics

Animal experiments were conducted according to the guidelines of the Animal Care (Okayama University Regulation No. 6 of 2008) and Use Committee of Okayama University. Animal experimentation was approved by Monash University Animal Experimentation Ethics Committee and was performed in accordance with Australian National Health and Medical Research Council Guidelines on Ethics in Animal Experimentation.

Reviewer #1 (Public review): https://doi.org/10.7554/eLife.99936.4.sa1
Reviewer #2 (Public review): https://doi.org/10.7554/eLife.99936.4.sa2
Reviewer #3 (Public review): https://doi.org/10.7554/eLife.99936.4.sa3
Author response https://doi.org/10.7554/eLife.99936.4.sa4

## Additional files

### Supplementary files
MDAR checklist

### Data availability
All data generated or analysed during this study are included in the manuscript and supporting files; source data files have been provided for *Figures 1, 2 and 4*.

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
