## [Editor Report · eLife Assessment]

This **important** study investigates the role of Drp1 in early embryo development. The authors have addressed most of the original comments and the work now presents **convincing** evidence on how this protein influences mitochondrial localization and partitioning during the first embryonic divisions. The research employs the Trim-Away technique to eliminate Drp1 in zygotes, revealing critical insights into mitochondrial clustering, spindle formation, and embryonic development.

---

## [Referee Report · Reviewer #1 (Public review)]

Summary:

Gekko, Nomura et al., show that Drp1 elimination in zygotes using the Trim-Away technique leads to mitochondrial clustering and uneven mitochondrial partitioning during the first embryonic cleavage, resulting in embryonic arrest. They monitor organellar localization and partitioning using specific targeted fluorophores. They also describe the effects of mitochondrial clustering in spindle formation and the detrimental effect of uneven mitochondrial partitioning to daughter cells.

Strengths:

The authors have gathered solid evidence for the uneven segregation of mitochondria upon Drp1 depletion through different means: mitochondrial labelling, ATP labelling and mtDNA copy number assessment in each daughter cell. Authors have also characterised the defects in cleavage mitotic spindles upon Drp1 loss

Weaknesses:

This study convincingly describes the phenotype seen upon Drp1 loss. Further studies should be conducted to elucidate the mechanism by which Drp1 ensures even mitochondrial partitioning during the first embryonic cleavage.

---

## [Referee Report · Reviewer #2 (Public review)]

Gekko et al investigate the impact of perturbing mitochondrial during early embryo development, through modulation of the mitochondrial fission protein Drp1 using Trim-Away technology. They aimed to validate a role for mitochondrial dynamics in modulating chromosomal segregation, mitochondrial inheritance and embryo development and achieve this through the examination of mitochondrial and endoplasmic reticulum distribution, as well as actin filament involvement, using targeted plasmids, molecular probes and TEM in pronuclear stage embryos through the first cleavages divisions. Drp1 deletion perturbed mitochondrial distribution, leading to asymmetric partitioning of mitochondria to the 2-cell stage embryo, prevented appropriate chromosomal segregation and culminated in embryo arrest. Resultant 2-cell embryos displayed altered ATP, mtDNA and calcium levels. Microinjection of Drp1 mRNA partially rescued embryo development. A role for actin filaments in mitochondrial inheritance is described, however the actin-based motor Myo19 does not appear to contribute.

Overall, this study builds upon their previous work and provides further support for a role of mitochondrial dynamics in mediating chromosomal segregation and mitochondrial inheritance. In particular, Drp1 is required for the redistribution of mitochondria to support symmetric partitioning and ongoing development.

Strengths:

The study is well designed, the methods are appropriate and the results are clearly presented. The findings are nicely summarised in a schematic.

The addition of further quantification, including mitochondrial cluster size, elongation/aspect ratio and ROS, as requested by the reviewers, has provided further evidence for the impact of Drp1 depletion on mitochondrial morphology and function.

Understanding the role of mitochondria in binucleation and mitochondrial inheritance is of clinical relevance for patients undergoing infertility treatment, particularly those undergoing mitochondrial replacement therapy.

Weaknesses:

The only remaining weakness is that the authors have not undertaken additional experiments to clarify any role for mitochondrial transport following Drp1 depletion.

---

## [Referee Report · Reviewer #3 (Public review)]

Why mitochondria are finely maintained in the female germ cell (oocyte), zygotes, and preimplantation embryos? Mitochondrial fusion seems beneficial in somatic cells to compensate for unhealthy mitochondria, for example, mitochondria with mutated mtDNA that potentially defuel the respiratory activity if accumulated above a certain threshold. However, in the germ cells, it may rather increase the risk of transmitting mutated mtDNA to the next generation. Also, finely maintained mitochondria would also be beneficial for efficient removal when damaged, as the authors briefly discussed. Due in part to the limited suitable model, physiological role of mitochondrial fission in embryos were obscure. In this study, authors demonstrated that mitochondrial fission prevents multiple adverse outcomes, especially including the aberrant demixing of parental genome (a clinical phenotype of human embryos) in zygotic stage. Thus, this study would be also of clinical importance that could contribute by proposing a novel mechanism.

The authors have adequately indicated the limitations at each of the specific points. The revisions the authors made have consolidated their conclusion, thus still, making this study an excellent one.

---

## [Author Response]

The following is the authors’ response to the previous reviews

**Reviewer #1 (Recommendations for the authors):**
The authors have taken into consideration and addressed all my previous comments.This referee has one major concern remaining: although the authors have refined their analysis of mitochondrial morphology, my concern regarding the characterization of mitochondria in Drp1-depleted zygotes as "elongated" persists.

Taking into account this reviewers' comment, the following description has been changed. Line 256-257: “Quantification of the aspect ratio (major axis/minor axis) suggests that mitochondria are significantly elongated in Drp1-depleted embryos" to “The mean aspect ratio (major axis/minor axis) increased slightly from 1.36 in control to 1.66 in Drp1-depleted embryos ."

(1) The morphological analysis of mitochondria reveals that both axes increase in length. Yet, the aspect ratio it is virtually unchanged, at least in biologically relevant terms, if not statistically.- Please calculate and represent mitochondrial aspect ratio as major axis/minor axis in fig 2M.- Could the authors also display individual data points in the graphs of Figure 2 K, L and M?

We have revised the graph display format in accordance with the reviewer's suggestions.

(2) The authors provide PMID: 25264261 as an example, yet mitochondria in PMID: 35704569 are apparently elongated. Judging by the authors discussion about the differences between these two studies, it would be enriching to comment, in the discussion of the manuscript, on the differences in morphology and to the reason why these might ariseThis referee believes that the unconventional mitochondrial morphology upon fission inhibition, reported here, enhances the relevance of the study and raises questions that could promote novel research lines, if thoroughly discussed in the manuscript.

Thank you for your insightful suggestion. However, since the latter paper (PMID: 35704569) lacks EM images, it would be difficult to accurately assess the elongation. Thus, we would like to reconsider the mitochondrial morphological changes in zygotes caused by Drp1 deletion levels based on the results of future research.

Minor(1) Labels for the staining used are missing in figure 1-figure supplement 1(2) Line 218. Could the intended sentence be:"Live imaging of mitochondria (mt-GFP) and chromosomes (H2B-mCherry) in Myo19 depleted zygotes shows symmetric distribution and partitioning of mitochondria during the first embryonic cleavage (Figure 1-figure supplement 2A, 2B; Figure 1-Video 2)."(3) Figure 2M: Please calculate and represent mitochondrial aspect ratio as major axis/minor axis.(4) Include a label with the experimental condition in figure 1 fig supp 2.(5) Line 592: missing reference.

Thank you for your careful correction. We have corrected all the points the reviewer pointed out in the revised version.

**Reviewer #2 (Recommendations for the authors):**
The authors have sufficiently revised the manuscript to accommodate the majority of suggestions provided by myself and the other reviewers. While it would have been useful to see further clarity around mitochondrial transport, the data presented provide valuable insight into the role of a mitochondrial dynamics regulator in mediating the first mitosis event in embryo development.

We thank again reviewer 2 for the helpful comment. We would like to address the issue of (aggregated) mitochondrial transport, including analysis methods, as a future challenge.

**Reviewer #3 (Recommendations for the authors):**
After reading through the comments of other reviewers, what authors could potentially improve their manuscript had been largely summarized in three following points.(1) Authors would better clarify whether a loss of Drp1 contributes to the chromosome segregation defects directly (e.g. checking SAC-like activity) or indirectly (aggregated mitochondria became physically obstacle; maybe in part getting the cytoskeleton involved).(2) Although the level of Myo19 may not be so high (given the low level of TRAK2 in oocytes: Lee et al. PNAS 2024, PMID 38917013), authors would better further clarify the effect of Myo19-Trim with timelapse (e.g. EB3-GFP/Mt-DsRed) and EM analysis (detailed mitochondrial architecture).(3) Authors would better clarify phenotypic heterogeneity/variety regarding the degree of alteration in mitochondrial morphology/ architecture dependent on the levels of Drp1 loss with detailed quantification of EM images to address why aggregation of mitochondria in Drp1-/- parthenote (possibly, more likely Drp1 protein-free) looks different/weaker than Trim-awayed one. Employment of the parthenotes of Trim-awayed MII oocytes might also complement the further discussion.The revised preprinted have addressed all the points described above. Authors have also adequately indicated the limitations at each of the specific points. Revisions authors made have consolidated their conclusion, thus still, making this study an excellent one.The only remaining weakness is that the authors have not undertaken additional experiments to clarify any role for mitochondrial transport following Drp1 depletion.

We thank again reviewer 3 for the insightful comments. We would like to address the comments you have raised (points that were unclear in this study) as issues for future study.